# Change in the Nature of ZSM-5 Zeolite Depending on the Type of Metal Adsorbent—The Analysis of DOS and Orbitals for Iron Species

**DOI:** 10.3390/ijms24043374

**Published:** 2023-02-08

**Authors:** Izabela Kurzydym, Alberto Garbujo, Pierdomenico Biasi, Izabela Czekaj

**Affiliations:** 1Department of Organic Chemistry and Technology, Faculty of Chemical Engineering and Technology, Cracow University of Technology, Warszawska 24, 31-155 Cracow, Poland; 2Casale SA, Via Pocobelli 6, CH-6900 Lugano, Switzerland

**Keywords:** ZSM-5, DFT, DOS, orbitals, zeolite, iron

## Abstract

Transition-metal-modified zeolites have recently gained the greatest interest among scientists. Ab initio calculations within the density functional theory were used. The exchange and correlation functional was approximated with the Perdew–Burke–Ernzerhof (PBE) functional. Cluster models of ZSM-5 (Al_2_Si_18_O_53_H_26_) zeolites were used with Fe particles adsorbed above aluminum. The adsorption of three iron adsorbates inside the pores of the ZSM-5 zeolite—Fe, FeO and FeOH—was carried out with different arrangements of aluminum atoms in the zeolite structure. The DOS diagram and the HOMO, SOMO and LUMO molecular orbitals for these systems were analyzed. It has been shown that depending on the adsorbate and the position of aluminum atoms in the pore structure of the zeolite, the systems can be described as insulators or conductors, which significantly affects their activity. The main aim of the research was to understand the behavior of these types of systems in order to select the most efficient one for a catalytic reaction.

## 1. Introduction

For heterogeneous catalysis, the use of ZSM-5 as a catalyst has been an issue engaging researchers over the past few decades [1,2,3,4]. The zeolite structures of ZSM-5 containing iron atoms were shown to have good catalytic activity in a number of reactions [5]. The most relevant aspect regarding research on the Fe-ZSM-5 zeolites is to understand their activity. Especially important are the Fe adsorbates that can exist as mono-, bi- or oligomers in the pores and also as oxides or different species [6,7].

It should be noted that the Fe-ZSM-5 catalysts, in addition to their activity in oxidation reactions, also perform a promising role in N_2_O decomposition reactions, as confirmed by both experimental and theoretical studies [8,9,10].

The presence of isolated iron adsorbates in the ZSM-5 zeolite is characterized when, for example, the ion exchange solution is very dilute [11,12,13]. However, these adsorbates show high activity in catalytic reactions. Therefore, the knowledge of the nature and properties, as well as the possibility of regulating the formation of active centers for the type of structures, are very important for a possible use of such catalysts in industrial processes.

The analysis of molecular orbitals and DOS (density of states) and PDOS (partial density of states) diagrams is very important for different types of catalytic systems [14,15]. The most important application of DOS diagrams is to show the MO (molecular orbital) composition. On the other hand, the analysis of the energy gap between the highest-energy filled orbital and the lowest-energy unfilled orbital allows the determination of the electron transport properties of molecules [16]. The energy difference between HOMO/SOMO (highest occupied molecular orbital/singly occupied molecular orbital) and LUMO (lowest unoccupied molecular orbital) gives the band gap energy, which plays an important role in determining the chemical stability and reactivity of a molecule [17]. The greater the difference between these terminal orbitals, the more stable a system is.

In attempting to investigate reaction mechanisms, the use of computational methods such as density functional theory (DFT) is proving to be very powerful [18]. The progress that has been made in the field of DFT calculations for surface processes is a very significant development that allows the design of catalysts for a variety of reactions [19,20]. Nowadays, it is only necessary to select the right method for the material and the calculations will be fast, adequate for reality and even allowing multi-atom systems to be studied [21,22]. Calculations can also provide information about the interaction energies between the molecules participating in the process and the metals on the surface of the catalyst, as well as the reactivity of transition metals [23]. The reaction with a heterogeneous catalyst proceeds schematically in three steps: (1) adsorption of molecules on the surface of the catalyst, (2) reaction with the participation of the catalyst, (3) desorption of the resulting products [24]. It should therefore be remembered that it is important to analyze both the adsorption properties of the substrates and the desorption properties of the products. This is why the catalyst should exhibit the so-called ‘optimum binding force’ for the molecules involved in the process, which is called Sabatier’s principle [25].

Zeolites of the MFI type (ZSM-5) can be used in many industrial processes such as hydrocarbon upgrading, isomerization, oligomerization or cracking processes [26,27]. In recent decades, zeolites, including ZSM-5, have also been considered as catalysts for valorization processes of alternative feedstocks such as natural gas [28] or biomass [29]. Understanding the different catalyst structures and reaction mechanisms allows the optimization and improvement of the heterogeneous catalyst and, consequently, the entire industrial process [30]. Zeolites can show different chemical compositions depending on the surface area being considered. The different geometrical structure can have a significant influence on the selectivity of the catalytic process [31,32,33]. In addition, in order to understand the functioning of the catalyst under the dynamic conditions of the real process, it is necessary to consider many possibilities of instantaneous modifications of the geometric and electronic structure of the active centers inside the zeolite. This variability is the reason for continuous research and identification of active sites in zeolites and studying their electronic structure.

This paper presents an analysis of the adsorption energy of iron adsorbates in the ZSM-5 zeolite pore and detailed characterization of their electronic structure. A detailed analysis of the bond lengths, bond orders and ionicity of the active center of the systems obtained was carried out. Further, HOMO, SOMO and LUMO orbitals for these structures were presented, with both alpha and beta orbitals due to the absence of symmetry in the system. In addition, PDOS diagrams for the individual atom types in the system are also presented. Finally, the ionicity of the atoms is illustrated based on the Mulliken population analysis. Such a detailed study of the electronic structure of catalytic systems makes it possible to predict the behavior of a selected catalyst in the catalytic process.

## 2. Results and Discussion

Figure 1 shows a pore of the ZSM-5 zeolite with adsorbed deposits—iron (Figure 1a), iron oxide (Figure 1b) and iron hydroxide (Figure 1c).

The adsorption energies of the individual iron adsorbates are presented below the structures. We can see that in all cases the adsorption process is endothermic, and extra energy is needed to form the desired structure. In addition, we can conclude that the most stable system is the one with adsorbed iron hydroxide (adsorption energy: 0.58 eV). In contrast, the ZSM-5 zeolite pore with adsorbed iron oxide is the most difficult to obtain. This structure is the least stable system (adsorption energy: 1.71 eV).

In the secondary analyses of the obtained structures, bond lengths, bond orders and ionicity were carried out for all systems (Figure 2). The analysis focused on a region containing the active centers (aluminum atoms) and the metallic molecules adsorbed upon them.

In the case of iron hydroxide, it only bonds to the zeolite structure via oxygen atoms (Figure 2c), while in the case of adsorption of an iron atom (Figure 2a) and iron oxide (Figure 2b), the iron forms an additional bond with an aluminum atom. However, this bond is very weak and its order is 0.04 for iron and 0.03 for iron oxide. On the other hand, the lengths and orders of the iron–oxygen bonds differ slightly from one structure to another, but these changes are not significant. Similarly, the ionicity of oxides bonding with iron is very similar for all systems. However, a rather significant change can be seen in the ionicity of the iron atom. As the structure of the adsorbate in the zeolite pore expands, the iron atom ionicity increases. Hence, for a single iron atom it is 0.88, for iron oxide it is 1.02 and for iron hydroxide it is 1.18. This may in turn impact the fact that the iron in the last case does not bond with the positive aluminum atom (0.76) due to a relatively small difference in the ionicity of these atoms. The difference between the ionicity of the oxygen atom in the iron oxide and the oxygen atom in iron hydroxide is also very significant. In the first case, the ionicity of the oxygen atom is −0.37; in the second case it is −0.78. In addition, differences in the order and bond length of the oxygen with iron are observed. In the ZSM-5 structure with FeO, the bond length is 1.60 Å and the bond order is 1.92, but in the ZSM-5 structure with FeOH the bond length is longer and is 1.75 Å and the bond order is smaller and is 1.01. This is influenced by the hydrogen atom, which is additionally attached to the oxygen atom in the latter structure.

The analysis of HOMO, SOMO and LUMO orbitals and DOS (density of states) diagrams for individual systems was also performed (Figure 3, Figure 4, Figure 5, Figure 6, Figure 7 and Figure 8). Due to the fact that most structures required calculations of higher multiplicities (the excited state allowed for lowering the energy of the system and its stabilization), for the structure with Fe the multiplicity is 5, for FeO it is 3 and for FeOH it is 6. The analysis presented HOMO (the highest-energy orbital fully filled), SOMO (orbitals filled singly between HOMO and LUMO) and LUMO (the lowest-energy unfilled orbital) orbitals. In addition, the α and β orbitals are also shown due to the absence of symmetry in the system.

Firstly, the pore orbitals of the ZSM-5 zeolite with a single iron atom were visualized (Figure 3).

First of all, there has been some minor energy switching. The first SOMO beta orbital has a lower energy than the HOMO beta orbital. In the case of both the alpha HOMO orbital and the alpha SOMO orbital, the orbital is located opposite to the iron silicon atoms and oxygen atoms in the zeolite pore. In contrast, the beta HOMO and SOMO orbitals are located mainly on iron. The exception is the first SOMO orbital, which also contains a fragment of the orbital on the zeolite pore. Perhaps this contributes to the lower energy of this orbital compared to the HOMO value. In the case of the LUMO orbital, both alpha and beta are located on iron (in the beta orbital, an additional part of the orbital includes aluminum). The localization of the LUMO orbital on iron may suggest the ease with which this system can adsorb molecules onto this type of structure and thus the system can be used as a catalyst for the reaction.

The gap between the last alpha orbital of SOMO and the first alpha orbital of LUMO can tell us whether the system will be either a conductor or an insulator. The energy difference between these orbitals is 1.36 eV; thus, the system will behave like an insulator.

The next analysis is the analysis of a DOS diagram. A diagram for the individual partial DOS for each atom—Si, O, H and Fe—as well as for the total DOS is presented (Figure 4). In addition, the diagram shows the location of the highest-energy SOMO alpha orbital and the lowest-energy LUMO alpha orbital. The DOS diagram shown is the result of summing the densities of the alpha and beta orbitals. Due to the divergence of the energies of the alpha and beta matrix orbitals (which are presented in Figure 3 and subsequent figures showing the orbitals), additional peaks appear between the energies of the alpha SOMO and LUMO orbitals after DOS summation, which correspond to the energy of the beta orbitals.

In the gap between the SOMO and LUMO orbitals, two peaks from oxygen and iron atoms are present. The gap between the SOMO and LUMO suggests an insulating character of this system. The DOS suggests the activity of iron and oxygen in further processes where the system can be used as a catalyst. In addition, the distance between these orbitals suggests that the system is relatively stable. It is also important to note that the most intense density of states is characterized by the oxygen atoms.

The next system analyzed in terms of the HOMO, SOMO and LUMO orbitals was the ZSM-5 zeolite pore with an iron oxide monomer deposited in the ZSM-5 pore (Figure 5).

The last alpha orbital of SOMO and the first alpha orbital of LUMO have exactly the same energy and they have a very similar structure. The last two beta orbitals show a certain difference, but this is also marginal. This indicates the instability of the system, which is also confirmed by the amount of energy required to form this structure. For the alpha orbitals, HOMO is located on the iron oxide, which may suggest a difficulty in applying this arrangement for catalytic processes. The alpha orbital appears only on the zeolite pore, and then splits partially onto the zeolite pore and the iron oxide. In the case of beta orbitals, these are concentrated near the iron oxide. A lower number of SOMO orbitals compared to the previous structure shows that in this case the multiplicity is lower and, consequently, the number of the excited states is lower as well.

A DOS diagram for the oxidized iron monomer was analyzed next (Figure 6).

As can be seen, the plotted alpha orbitals of the SOMO and LUMO show a complete absence of a gap between them, which suggests a conductive character of this system.

The whole analysis of the electronic structure of the FeO monomer inside the ZSM-5 pore suggests that it will be unstable and that it will be difficult to predict its behavior in any further catalytic process for which it might be used. In this case, as in the previous one, the oxygen atoms show the highest intensity of atomic density.

The last HOMO, SUMO and LUMO analyses were performed for iron hydroxide inside the ZSM-5 zeolite pore (Figure 7). They show the highest multiplicity and therefore the most SOMO-type orbitals.

A significant difference between the alpha and beta orbitals of the FeOH/ZSM-5 system is visible. In this case, all of the alpha orbitals (HOMO, SOMO and LUMO) are located on the zeolite pore, some only near the iron hydroxide. Additionally, the LUMO orbital localizes on the active center, which is aluminum, so this provides a basis for a possible use of the system in a further catalytic process. As for the beta orbitals, most orbitals are located on the iron hydroxide but additionally show localization on the zeolite pore. The difference between the last alpha orbital of SOMO and the alpha orbital of LUMO is also quite promising here. It amounts to 2.17 eV and shows that this system is the most stable of all those presented in the paper.

As for the previous cases, a DOS diagram was also presented (Figure 8).

The gap between the last orbitals is significant here. In addition, it is obvious that in this gap the intensity of the density of states is shown by the oxygen atoms (blue peak) and, to a small extent, by the iron atom (brown peak). As mentioned earlier, this is a promising signal for further use of the system. The highest intensity of the density of states is shown by the oxygen atoms. The gap between the SOMO and LUMO suggests the insulating nature of the FeOH monomer, like in the case of the Fe monomer.

The last and very interesting analysis consisted of performing calculations to visualize the atomic charges derived from the Mulliken populations used in the calculations (Figure 9).

The red color indicates a negative charge of the atom. As can be seen in the figure, in all cases the oxygen atoms show the negative charge. The blue color is a positive charge shown by the atoms of silicon, aluminum, iron and hydrogen. In addition, this visualization allows the size of the charges to be represented according to the size of a sphere that corresponds to each atom. Here, we can clearly see that the sphere corresponding to iron (violet blue), i.e., the one inside the pore, increases in correspondence with the changes in the increase in the positive charge on iron as described earlier. In addition, we can also see the difference in the oxygen charge in iron oxide and iron hydrate. In the first case, the charge is negative, but it is merely −0.37, while in the case of the second system, the sphere corresponding to the oxygen from the OH group is almost twice as large, as it represents a charge of −0.77. The charge localized on iron increases as follows: Fe (0.88) < Fe-O (1.02) < Fe-OH (1.18). The positive charge localized on the whole iron monomer is similar for Fe (0.88) and Fe-OH (0.86), but is lower on Fe-O (0.65). The visualization of the charges of the Mulliken populations gives us an interesting picture of the distribution and changes in the charges without going into their numerical values.

## 3. Materials and Methods

### 3.1. Computational Details

The ab initio density functional theory (DFT) method was used to calculate the electron structure of selected systems using the StoBe software [34]. The exchange and correlation functional was approximated with the Perdew–Burke–Ernzerhof (PBE) functional [21,22]. It was used to account for the electron exchange and correlation. By linear combinations of atomic orbitals (LCAOs) and using conventional Gaussian basis sets for atoms, the Kohn–Sham orbitals were presented [35]. Due to the peculiarities of transition metals (e.g., iron used), we use calculations for higher multiplicities (the correct multiplicity is considered to be the one for which the calculated total energy of the system reaches a minimum). Thus, from a computational point of view, in the calculation program used, the alpha and beta electron matrices are calculated separately (compared to calculations with the lowest multiplicity of one with alpha and beta paired electrons, where only the alpha matrix is calculated and copied for the beta electron matrix). This has implications in the energies of the individual levels of alpha and beta orbitals for high-spin systems.

Mulliken populations [36] and the Mayer bond order factors [37,38] were used to analyze the electron structure of the clusters. Molecular orbitals were also calculated for the systems and visualization was presented using the Molekel program [39]. The density of state (DOS) analysis is managed based on Mulliken populations, including all atoms (and also artificial hydroxyl groups at the periphery of used clusters) and all orbitals (alpha and beta).

The double-zeta valence polarization (DZVP) functional bases were used for the orbital basis sets Si and Al (6321/521/1), Fe (63321/531/311), O and N (621/41/1) and H (41). Additionally, their auxiliary functional bases were used to adjust the electron density and the exchange potential of the correlation of individual atoms: Si and Al (5,4;5,4), Fe (5,5;5,5), O and N (4,3;4,3) and H (4,0;4,0).

### 3.2. Geometrical Models

The structure of the ZSM-5 zeolite was taken from the Database of Zeolite Structure [40]. A single crystal unit cell contains 201 atoms, and the orthorhombic phase of the ZSM-5 zeolite framework type is characterized by the space group Pnma (#62) with lattice constants: a = 20.0900 Å, b = 19.7380 Å and c = 13.1420 Å.

To create the cluster required for the calculations, a fragment of a crystal-containing pore of the ZSM-5 zeolite with two aluminum and eighteen silicon atoms (Al_2_Si_18_O_53_H_26_) was cut out (Figure 10). The broken bonds were saturated with a point charge, which was represented by a hydrogen atom. The hydrogen was placed at a distance of 0.97 Å in a direction consistent with that presented by the broken Si-O bond. This structure was successfully used in earlier studies [41], where iron species with different saturation were also used to simulate different reaction conditions in real industrial conditions (oxygen/water presence or deficiency). In our current studies, we are using the knowledge of the localization of iron centers in the ZSM-5 network from these earlier studies to successively reduce the possibility of localization of iron particles in the zeolite network. The iron stabilizes inside the zeolite pore near the aluminum centers of the zeolite frame. Our previous modeling of this system indicates that this is the only case where iron particles stabilize within the zeolite pore. The modeling of adsorption near the lattice silicon atoms showed instability of the iron particles.

In order to analyze the changes in the nature of the ZSM-5 zeolite during dynamic reaction conditions, the adsorption of different types of iron adsorbates—a single iron atom, iron oxide and iron with an OH group—was carried out.

## 4. Conclusions

To summarize the presented analyses, we conclude the following:The most stable structure is the structure with the hydroxyl group on the iron.The formation of iron monomers is highly endothermic and they increase in the following order: Fe-OH < Fe << FeO.The difference in the HOMO/SOMO and LUMO orbital energy gives us information about a conductive nature of the iron species inside the zeolite. This character changes from an insulator (Fe/ZSM-5) to a conductor (FeO/ZSM-5) to an insulator (FeOH/ZSM-5), which will have an important effect on the reactivity of the whole system during a catalytic reaction.From the molecular orbital visualization, it can be seen that the catalyst with iron absorbed inside the ZSM-5 pore is likely to be the most active—the LUMO orbital is located directly on the iron.The ionicity of iron increases after bonding with oxygen or an OH group. It starts at 0.88 for the iron atom, then amounts to 1.02 for iron in the oxide and to 1.18 for iron with the OH group.

The theoretical analysis of the possible iron conformations present in real catalysts makes it possible to predict the behavior of the active centers under specific process conditions for industrial processes such as, for example, DeNOx and DeN_2_O processes. The identification of the most active form of iron on the catalyst makes it possible to adapt the method for obtaining the catalyst in its most efficient state, which significantly reduces the cost of the experimental studies and shortens their implementation time. The presented results of the study of the electronic structure of various forms of iron in the zeolite network also allow for the analysis of the variability of the structure under the dynamic conditions of a real industrial chemical process occurring under varying conditions of the reaction environment (for example, under aerobic or anaerobic conditions or in the presence of moisture). In the future, other transition metals in the zeolite frame, such as Cu, Mn or Zn, also with their polimeric forms, both mono- and bimetallic, should be investigated using the same approach.

## Figures and Tables

**Figure 1 ijms-24-03374-f001:**
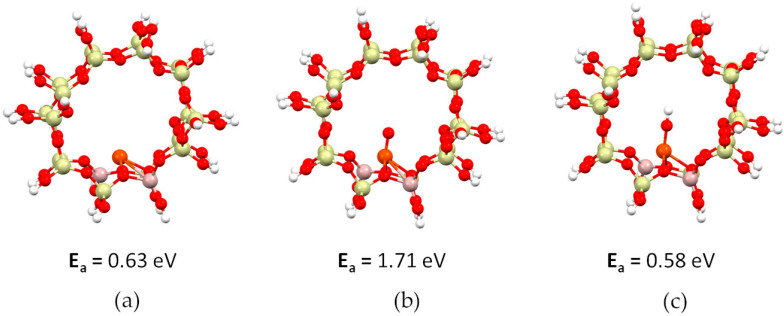
Structure with iron adsorbates inside the pore (energy of adsorption below the structure): (**a**) ZSM-5 with iron, (**b**) ZSM-5 with iron oxide, and (**c**) ZSM-5 with iron hydroxide. Atoms’ color: iron—orange, aluminum—pink, silicon—yellow, oxygen—red, hydrogen—white.

**Figure 2 ijms-24-03374-f002:**
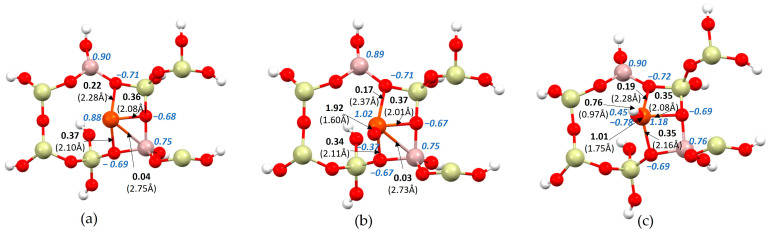
Fragment of structures with active iron adsorbates inside the pore: (**a**) ZSM-5 with iron, (**b**) ZSM-5 with iron oxide, and (**c**) ZSM-5 with iron hydroxide. Italic blue—ionicity, bold—bond order, bracket—bond length.

**Figure 3 ijms-24-03374-f003:**
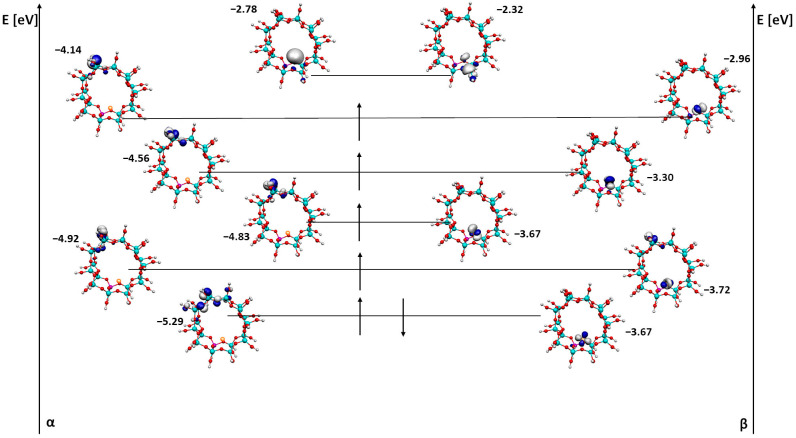
HOMO, SOMO and LUMO alpha and beta orbitals for ZSM-5 with Fe adsorbed next to the active site.

**Figure 4 ijms-24-03374-f004:**
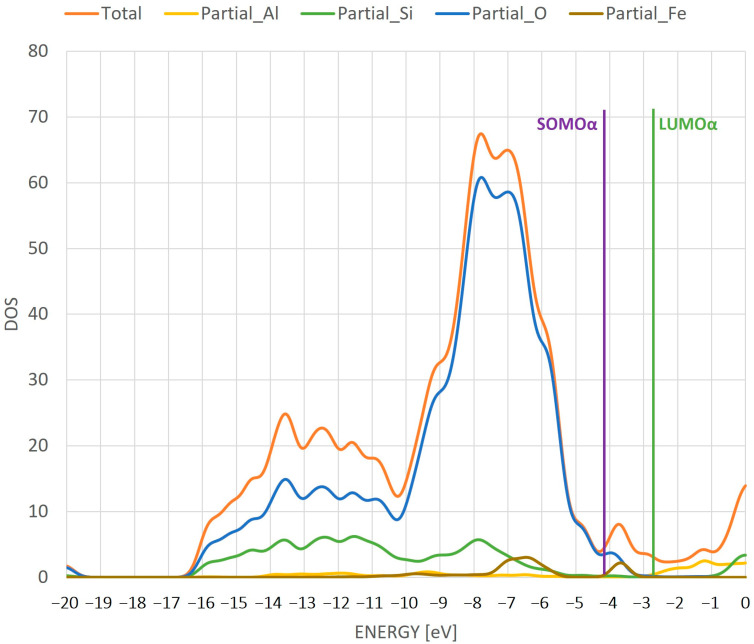
DOS diagram for individual atoms in the ZSM-5 structure with Fe adsorbed on the active site. The line in the diagram shows the position of SOMO and LUMO orbitals.

**Figure 5 ijms-24-03374-f005:**
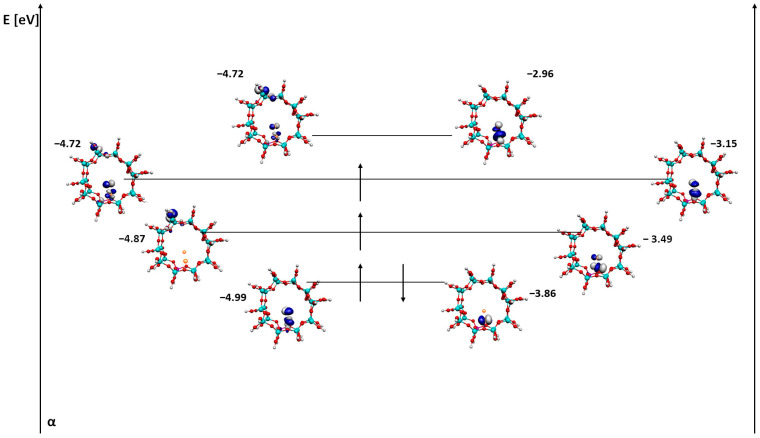
HOMO, SOMO and LUMO alpha and beta orbitals for ZSM-5 with FeO adsorbed next to the active site.

**Figure 6 ijms-24-03374-f006:**
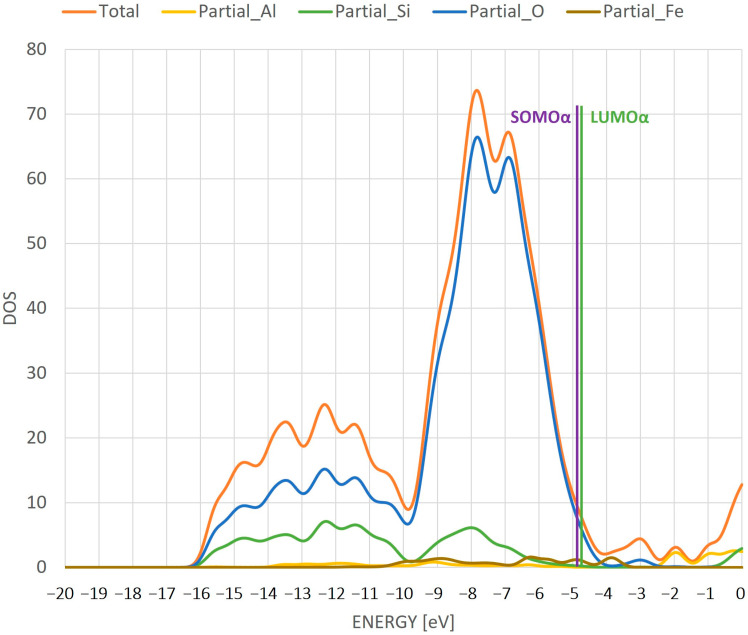
DOS diagram for individual atoms in the ZSM-5 structure with FeO adsorbed on the active site. The line in the diagram shows the position of SOMO and LUMO orbitals.

**Figure 7 ijms-24-03374-f007:**
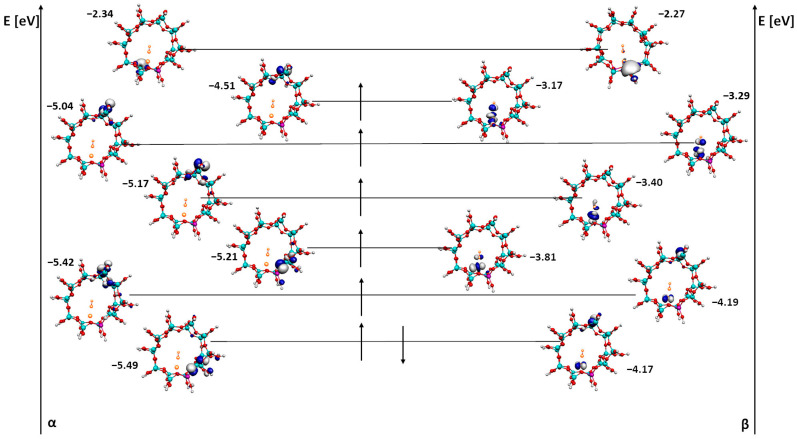
HOMO, SOMO and LUMO alpha and beta orbitals for ZSM-5 with FeOH adsorbed next to the active site.

**Figure 8 ijms-24-03374-f008:**
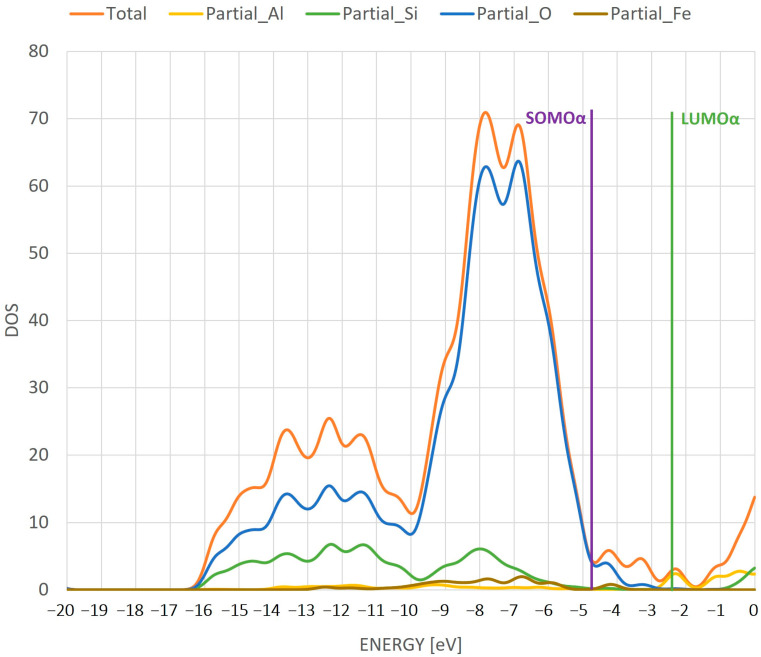
DOS diagram for individual atoms in the ZSM-5 structure with FeOH adsorbed on the active site. The line in the diagram shows the position of SOMO and LUMO orbitals.

**Figure 9 ijms-24-03374-f009:**
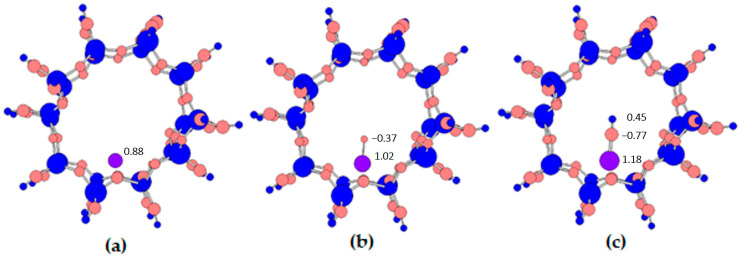
Visualized atomic charges derived from the Mulliken populations (blue—positive charge, red—negative charge): (**a**) ZSM-5 structure with Fe, (**b**) ZSM-5 structure with FeO, and (**c**) ZSM-5 structure with FeOH.

**Figure 10 ijms-24-03374-f010:**
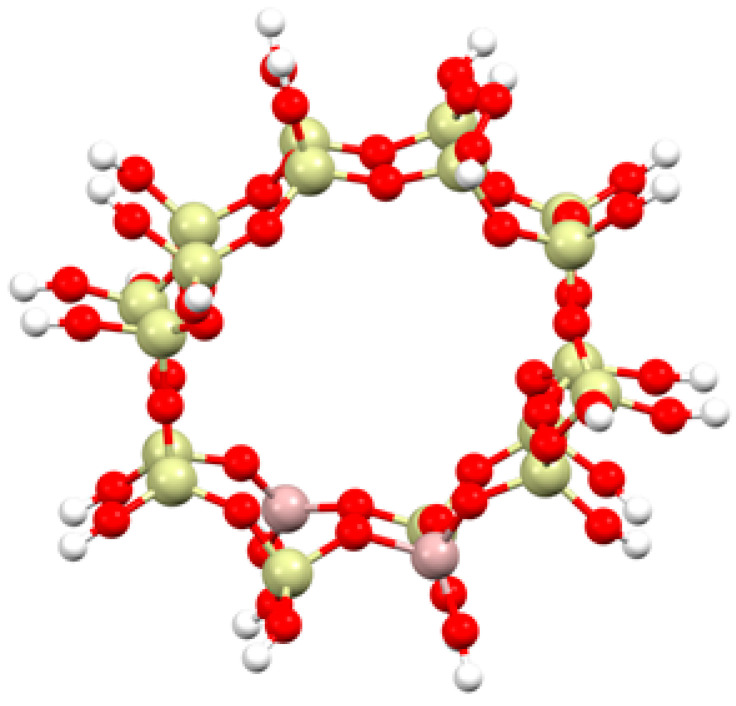
Structure used for the ZSM-5 zeolite (Al_2_Si_18_O_53_H_26_) calculations.

## Data Availability

Data are contained within the article.

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
