# Peer review of "Change in the Nature of ZSM-5 Zeolite Depending on the Type of Metal Adsorbent—The Analysis of DOS and Orbitals for Iron Species"

_ijms, 2023, doi:10.3390/ijms24043374_

Round 1

Reviewer 1 Report

In this work, the authors used DFT with the GGA functional PBE to calculate zeolites with different iron adsorbates. By analyzing the orbital energy, the density of state and the Mulliken charge population, the stability and the reactivity of different zeolites are compared. The reactivity information is helpful for understanding the catalysis reaction, which could attract attentions from both theoretical and experimental community. As such, the proposed article deserves to be published and the IJMS is certainly well targeted.

Before publication, the referee asks the authors to address the following questions and comments.

Details:

1. Page 1, line 38

The abbreviations “DOS” and “PDOS” are not defined before using.

2. Page 1, line 40

The abbreviation “MO” is not defined before using.

3. Page 1, line 43

The abbreviations “HOMO”, “LUMO” and “SOMO” are not defined before using.

4. Page 2, Figure 1

The legend for “colors- atom species” is needed. The resolution of this figure is low.

5. Page 3, line 96

“Due to the fact that most structures required calculations of higher multiplicities”

The multiplicity should be labels in the figure or stated here.

6. Page 5, Figure 4

The vertical lines for SOMO and LUMO are not at the position of the peaks of the DOS. Can the authors provide an explanation for this?

7. Page 5, Figure 4

There is a peak between the SOMO line and the LUMO line, what is the meaning for this peak?

8. Page 7, Figure 6

The vertical lines for SOMO and LUMO are not at any peak in the DOS diagram. Can the authors provide an explanation for this?

Author Response

Dear Reviewer,

We are very appreciate to Reviewer for essential comments.

We follow all points suggested by Reviewer and revised our manuscript accordingly.

With kind regards,

Izabela Czekaj

Reviewer 2 Report

The article presented for review, "Change in the nature of ZSM-5 zeolite depending on the type 2

of metal adsorbent - the analysis of DOS and orbitals for iron 3 species" is an extremely interesting paper. It touches on a topic that is currently being studied by many scientists. Each of the presented parts of the publication brings a very high scientific value. Each of the presented parts was developed very meticulously. The content provided by the authors is clear and readable for the recipient. 

I do not make major comments on the content, there are only minor editorial errors that do not affect the quality of the work:

Line 206 should be removed unnecessary period "4.1. . Computational details"

Have the modified forms of zeolite ZSM-5 presented in the paper been or will be used in any chemical processes?

Author Response

(The authors gave the same response as above.)
